# Factors associated with mortality in children under five years old hospitalized for Severe Acute Malnutrition in Limpopo province, South Africa, 2014-2018: A cross-sectional analytic study

**Fhatuwani Gavhi** [1,2]*, **Lazarus Kuonza**[1,2,3], **Alfred Musekiwa**[2], **Nkengafac Villyen Motaze**[1,4]

**1** Division of the National Health Laboratory Service, National Institute for Communicable Diseases, Johannesburg, South Africa, **2** School of Health Systems and Public Health, Faculty of Health Sciences, University of Pretoria, Pretoria, South Africa, **3** School of Public Health, Faculty of Health Sciences, University of Witwatersrand, Johannesburg, South Africa, **4** Department of Global Health, Faculty of Medicine and Health Sciences, Stellenbosch University, Cape Town, South Africa

* fhatuwanigavhi@yahoo.com

**Data Availability Statement:** All relevant data are within the manuscript Supporting Information files.

## Abstract

### Background

In South Africa, 30.9% of children under five years with Severe Acute Malnutrition (SAM) died in 2018. We aimed to identify factors associated with mortality among children under five years hospitalized with SAM in Limpopo province, South Africa.

### Methods

We conducted a cross-sectional study including children under five years admitted with SAM from 2014 to 2018 in public hospitals of Limpopo province. We extracted socio-demographic and clinical data from hospital records. We used logistic regression to identify factors associated with mortality.

### Findings

We included 956 children, 50.2% (480/956) male and 49.8% (476/956) female. The median age was 13 months (inter quartile range: 9–19 months). The overall SAM mortality over the study period was 25.9% (248/956). The most common complications were diarrhea, 63.8% (610/956), and lower respiratory tract infections (LRTIs), 42.4% (405/956). Factors associated with mortality included herbal medication use (adjusted Odds Ratio (aOR): 2.2, 95% Confidence Interval (CI): 1.4–3.5, p = 0.001), poor appetite (aOR: 2.7, 95% CI: 1.4–5.2, p = 0.003), Mid-upper circumference (MUAC) <11.5 cm (aOR: 3.0, 95% CI: 1.9–4.7, p<0.001), lower respiratory tract infections (LRTIs) (aOR: 1.6, 95% CI: 1.2–2.0, p<0.001), anemia (aOR: 2.5, 95% CI: 1.1–5.3, p = 0.021), hypoglycemia (aOR: 12.4, 95% CI: 7.1–21.8,

**Funding:** The author(s) received no specific funding for this work.

**Competing interests:** The authors have declared that no competing interests exist.

p<0.001) and human immunodeficiency virus (HIV) infection (aOR: 2.3, 95% CI: 1.6–3.3, p<0.001).

## Interpretation

Herbal medication use, poor appetite, LRTIs, anemia, hypoglycemia, and HIV infection were associated with mortality among children with SAM. These factors should guide management of children with SAM.

## Introduction

Severe Acute Malnutrition (SAM) is defined as a weight for height below -3 Z-scores of the median World Health Organization (WHO) growth standards, or visible severe wasting, or the presence of bilateral pitting edema in children under five years old [1]. A Mid-Upper Arm Circumference (MUAC) <11.5 cm is also indicative of SAM in children 6–59 months of age [1]. SAM is a life threatening condition, and has been associated with poverty, inadequate nutrient intake, lack of access to adequate health services, and concurrent diseases [1, 2]. Children with SAM have weakened immune system, are more susceptible to diseases and have an increased risk of mortality [3, 4].

SAM is an important public health problem and a major contributor to morbidity and mortality among children under five years worldwide [5]. Joint global estimates from the United Nations International Children's Emergency Fund (UNICEF), WHO, and the World Bank revealed that nearly 17 million children under five years old had SAM in 2018, with 4.4 million from Sub-Saharan Africa [5]. This report shows insufficient progress towards the 2025 targets of the World Health Assembly [6] and Sustainable Development Goal number three, addressing preventable deaths among children under five years [7].

Children with SAM require urgent treatment to prevent death since their weakened immune system makes them nine times more likely to die compared to their well-nourished counterparts [2, 3]. It is estimated that approximately one million children under five years with SAM die every year globally [2, 5]. The WHO developed guidelines for the management of SAM with the aim of preventing mortality. These management guidelines are currently the gold standard for managing SAM in hospitals. The guidelines recommend a ten-step protocol with three distinct phases; stabilization, rehabilitation, and discharge and follow-up [1]. With proper adherence to these guidelines, SAM mortality can be reduced to less than 10% in hospitalized children [1].

Despite proper adherence to the SAM management guidelines reported in certain studies in low- and middle-income countries, including South Africa, [8–10] mortality rates in Sub-Saharan Africa still surpass 40% among children under five years hospitalized with SAM [11, 12]. Mortality rates among children hospitalized with SAM were as high as 40.1% in Swaziland [11] and 46% in Zambia [12]. This demonstrates a gap in terms of implementing WHO SAM management standards. A number of studies conducted in Sub-Saharan Africa have identified contributors to high inpatient mortality among children with SAM; late presentations of cases at health facilities, inappropriate case management, co-morbidities such as Tuberculosis (TB), Human Immunodeficiency virus (HIV) infection, and complications such as diarrhoea anemia, hypoglycemia and lower respiratory tract infections (LRTIs) [13–16].

The District Health Barometer reported that 11 229 children under five years were treated for SAM from April 2017 to March 2018 in South Africa [17]. During this period, SAM was an

underlying cause of mortality in 30.9% of deaths among children under five years in South Africa [17]. An earlier report suggested that approximately 33% of deaths from 2011 to 2013 among children under five years in South Africa was associated with SAM [18]. In addition, a study conducted in two rural hospitals of Eastern Cape province among children hospitalized for SAM reported mortality in 24.4% of cases, which is much higher than the WHO acceptable standard (<10%) [19]. These high mortality figures suggest possible unexplored factors that contribute to mortality among children with SAM. While studies on the causes of SAM [20] and implementation of WHO SAM management guidelines [8, 10, 21] have been conducted in South Africa, few have examined factors for mortality. This study aimed to determine factors associated with mortality in children under five years hospitalized for SAM in public hospitals of Limpopo province.

## Materials and methods

### Study design and setting

We conducted a cross-sectional analytic study using hospital records of children under five years who were admitted with SAM in public hospitals of Limpopo province from 2014 to 2018. Limpopo province is situated in the northern-eastern part of South Africa with a population of 5,797,275 million [22]. Limpopo is the fifth most populated province in the country following Gauteng, KwaZulu-Natal, Eastern Cape, and Western Cape provinces [22]. Limpopo is divided into five health districts; Capricorn, Sekhukhune, Mopani, Vhembe, and Waterberg. We used a convenience sampling method and selected Sekhukhune and Waterberg districts for the study. We included seven hospitals, three regional hospitals and four district hospitals.

### Study size and sampling

We included all children under five years old admitted with SAM from 2014 to 2018 at the study sites and excluded children with missing hospital records.

### Operational definitions

Mortality: Any child under five years old admitted with SAM who died following admission to hospital.

Adequate feeding: Exclusive breastfeeding or exclusive formula feeding children aged 0 to 6 months. For children aged 7 to 59 months, breast milk or formula milk with additional food items.

Inadequate feeding: For children aged 0 to 6 months, breastfeeding or formula feeding with any additional food items or solid feeds only. For children aged 7 to 24 months, breastfeeding or formula feeding only, or solid feeds only. For children aged 25 to 59 months, complementary feeds that do not include a variety of food items to cover the child's nutritional needs.

Co-morbidity: Occurrence of one or more medical conditions prior to the diagnosis of SAM in an individual.

Complications of SAM: Any medical condition occurring in a child with SAM.

### Measurements

We reviewed the admission registers in the pediatric ward to identify children under five years admitted with SAM and identified their medical records using hospital numbers. We extracted information on outcome and exposure variables of interest from hospital records using a pre-designed data capture form. We further extracted information on treatment received and

patient outcomes. We audited completeness and clarity of information on the data capture form at the end of each day to ensure accuracy of information.

## Variables

Our outcome variable of interest was mortality amongst children admitted with SAM. Our exposure variables included socio-demographics and clinical data. Socio-demographic characteristics included age and sex of the children as well as age, sex, employment status and education level of the caregiver(s). Clinical data included appetite test results, immunization status, feeding practices and herbal medication use. We also extracted data on the previous admissions if applicable. We further extracted data on clinical examination; Z-scores, MUAC and edema. Where information on Z-scores was not available, we plotted growth charts using values for weight and height obtained from patient records. We also extracted information on co-morbidities, complications and treatment received (antibiotics, vitamin A supplements, rehydration solution, oxygen therapy, and feeding type).

## Data management and analysis

We used Microsoft Excel (2016) for data capture and cleaning. We imported the dataset into Stata (*Version 15*. StataCorp LLC, College Station, TX, United State of America, 2017) for analysis. We used descriptive statistics for socio-demographic and clinical characteristics. We presented numerical data using median, inter quartile range (IQR), mean and standard deviations (SD). We presented categorical variables using absolute numbers and percentages. We used frequency tables and bar charts to display data visually. We calculated SAM mortality as the total number of SAM deaths divided by the total number of children admitted with SAM. We represented trends in mortality over time using line graphs. We used univariable and multivariable logistic regression analysis to assess factors associated with mortality. We used a p-value of 0.2 as a cut-off value for inclusion in the multivariable logistic regression model. We used stepwise logistic regression approach and all variables that remained statistically significant at a cut-off p-value of 0.05 were included in the final model. We conducted multiple imputation by chained equations to account for missing data on appetite test and measurements of MUAC. We created 50 imputation data sets. We used robust standard errors in our final model to account for correlation within clusters (hospitals). We reported adjusted odds ratios (aORs) from the final model with the corresponding 95% confidence intervals (CIs) and p-values.

## Ethical considerations

The study was approved by the Faculty of Health Sciences Research Ethics Committee of the University of Pretoria (Ethic No 551/2018). The research ethics committee also waived the informed consent of the caregivers of the children who were included in the study (S1 File). We obtained approvals from the management of each included health facility.

## Results

A total of 1424 children were admitted with SAM from 2014 to 2018 in the selected hospitals (Fig 1). We excluded 32.9% (468/1424) children due to missing of hospital records. Therefore, 67.1% (956/1424) children were included in the study.

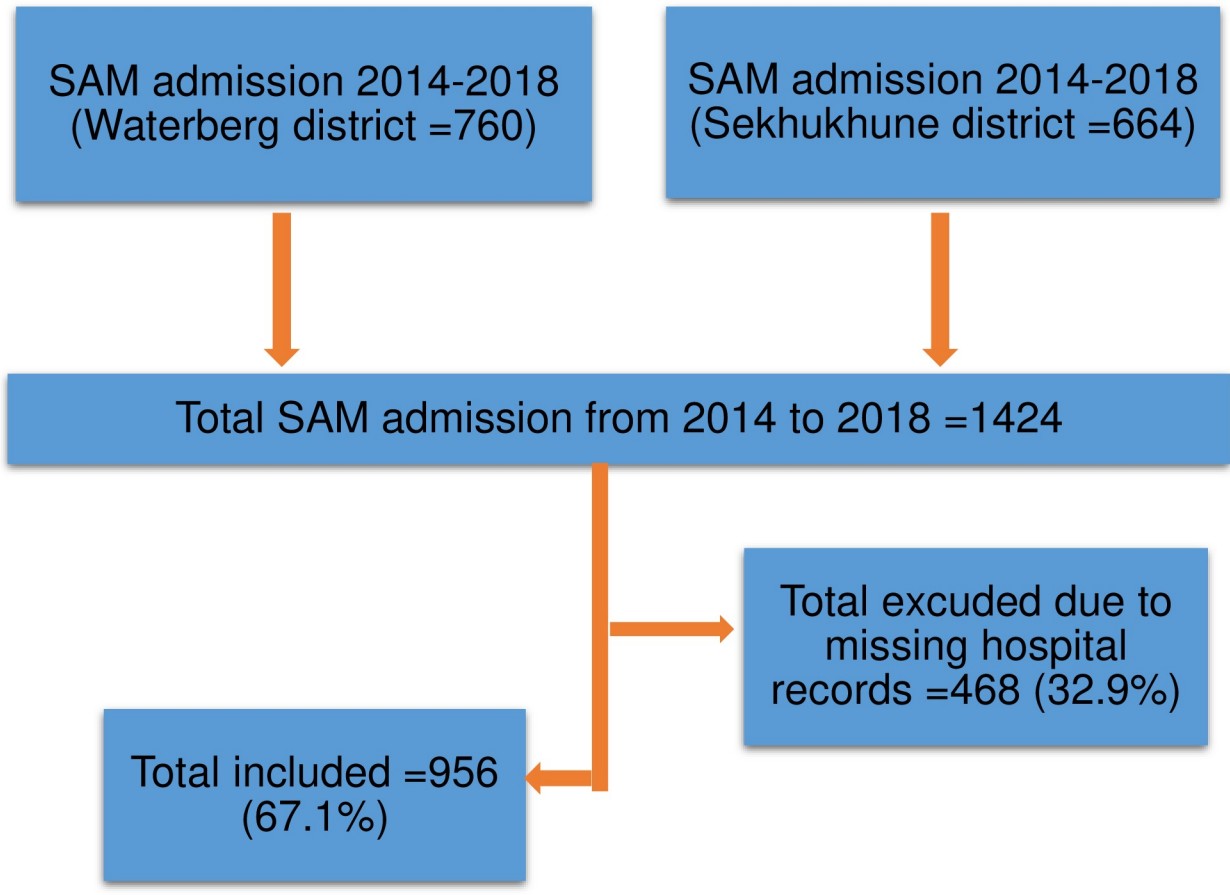

**Fig 1. Study flow diagram for sample selection in the two included districts, Limpopo province, 2014–2018.**

### Socio-demographic characteristics of children with SAM

The median age was 13 months (IQR: 9–19 months) and 73.8% (706/956) of the children were aged between seven and 24 months (Table 1). Just over half (50.2%) of the children were males and the majority (99.3%) of caregivers were females. Data on the ages of caregivers were available for 1.0% (10/956) of cases. The median age of the caregivers was 30 years (IQR: 25–33 years). We did not find data on the level of education and employment status of the caregivers.

**Table 1. Socio-demographic characteristics of children under five years admitted with SAM at the study sites, 2014–2018.**

| Variable (N = 956) | Categories | n (%) |
|---|---|---|
| Age of children (Months) | Median age (IQR) | 13 (9–19) |
|  | 0–6 | 120 (12.6) |
|  | 7–24 | 706 (73.8) |
|  | 25–59 | 130 (13.6) |
| Sex of children | Female | 476 (49.8) |
|  | Male | 480 (50.2) |
| Sex of caregivers | Female | 949 (99.3) |
|  | Male | 7 (0.7) |

IQR = Inter-quartile range

## Clinical characteristics of children with SAM

The majority of the children (84.6%) were admitted with SAM for the first time (Table 2). The mean MUAC was 11.3 cm (SD = 1.4) and 57.6% (551/956) of the children had Z-scores below -3 of the median WHO growth standards. About 68.4% (654/956) of the children had poor appetite and 61.5% (588/956) had non-edematous SAM on admission. About 62.1% (594/956) of the children were up to date with their immunization schedule and 17.6% (168/956) had a history of using herbal medicines. About 37.5% (45/120) of children, aged 0 to 6 months had adequate feeds before admission.

## Co-morbidities and complications of SAM

The attending clinicians made diagnoses of co-morbidities and complications of SAM during hospitalization. About 18.9% (181/956) of the children were HIV infected (Table 3). We did not collect data on receipt of antiretroviral treatment for HIV infected children. Diarrhoea and

**Table 2. Clinical characteristics of children under five years admitted with SAM at the study sites, 2014–2018.**

| Variable (N = 956) | Categories | n (%) |
|---|---|---|
| Admission type | New | 809 (84.6) |
| | Readmission | 147 (15.4) |
| MUAC (cm) | Mean MUAC (SD) | 11.3 (1.4) |
| | ≥11.5 | 278 (29.1) |
| | <11.5 | 483 (50.5) |
| | Unknown | 195 (20.4) |
| Weight for length Z-Scores | ≥ -3 | 307 (32.1) |
| | <-3 | 551 (57.6) |
| | Unknown | 98 (10.3) |
| Appetite test | Good | 209 (21.9) |
| | Poor | 654 (68.4) |
| | Unknown | 93 (9.7) |
| Oedema | Non-oedematous | 588 (61.5) |
| | Oedematous | 368 (38.5) |
| Immunization status | Up to date | 594 (62.1) |
| | Not up to date | 281 (29.4) |
| | Unknown | 81 (8.5) |
| History of herbal medication use | Yes | 168 (17.6) |
| | No | 788 (82.4) |
| Feeding history | | |
| 0 to 6 months (n = 120) | Adequate feeding | 45 (37.5) |
| | Inadequate feeding | 69 (57.5) |
| | Unknown | 6 (5.0) |
| 7 to 24 months (n = 706) | Adequate feeding | 54 (7.6) |
| | Inadequate feeding | 523 (74.1) |
| | Unknown | 129 (18.3) |
| 25 to 59 months (n = 130) | Adequate feeding | 81 (62.3) |
| | Inadequate feeding | 23 (17.7) |
| | Unknown | 26 (20.0) |

Unknown refers to the data that were not available on the records. MUAC = Mid-Upper Arm Circumference.
SD = Standard Deviation

**Table 3. Co-morbidities and complications in children under five years admitted with SAM, at the study sites, 2014–2018.**

| Characteristics (N = 956) | Categories | n (%) |
|---|---|---|
| **Co-morbidities** | | |
| HIV infection | Infected | 181 (18.9) |
| | Not infected | 775 (81.1) |
| TB | Infected | 127 (13.3) |
| | Not infected | 829 (86.7) |
| **Complications** | | |
| Malaria | No | 952 (99.6) |
| | Yes | 4 (0.4) |
| Diarrhoea | No | 346 (36.2) |
| | Yes | 610 (63.8) |
| Anaemia | No | 783 (81.9) |
| | Yes | 173 (18.1) |
| Hypoglycaemia | No | 860 (90.0) |
| | Yes | 96 (10.0) |
| LRTIs | No | 551 (57.6) |
| | Yes | 405 (42.4) |

TB = Tuberculosis. HIV = Human Immunodeficiency Virus. LRTIs = Lower respiratory tract infections

LRTIs were the most common complications occurring in 63.8% (610/956) and 42.4% (405/956) of cases respectively.

## Treatment received during admission

Antibiotics and rehydration therapy were administered to 85% (813/956) and 80.8% (772/956) of cases respectively (Table 4). Vitamin A supplementation was given to 64.6% (618/956) of cases and 14.4% (138/956) were fed using naso-gastric tubes.

## Admission outcomes

Of 956 children admitted with SAM from 2014 to 2018, 25.9% (248/956) died with 14.1% (35/248) of deaths occurring on the day of admission. About 74.1% (708/956) were alive on

**Table 4. Treatment given to children under five years admitted with SAM at the study sites, 2014–2018.**

| Characteristics (N = 956) | Categories | n (%) |
|---|---|---|
| Antibiotics use | Yes | 813 (85.0) |
| | Unknown | 143 (15.0) |
| Rehydration therapy | Yes | 772 (80.8) |
| | Unknown | 184 (19.3) |
| Vitamin A supplementation | Yes | 618 (64.6) |
| | Unknown | 338 (35.4) |
| Feeding type | Oral | 818 (85.6) |
| | Naso-gastric tube | 138 (14.4) |
| Oxygen therapy | Yes | 229 (24.0) |
| | No | 727 (76.1) |

Unknown refers to the data that were not available on the records

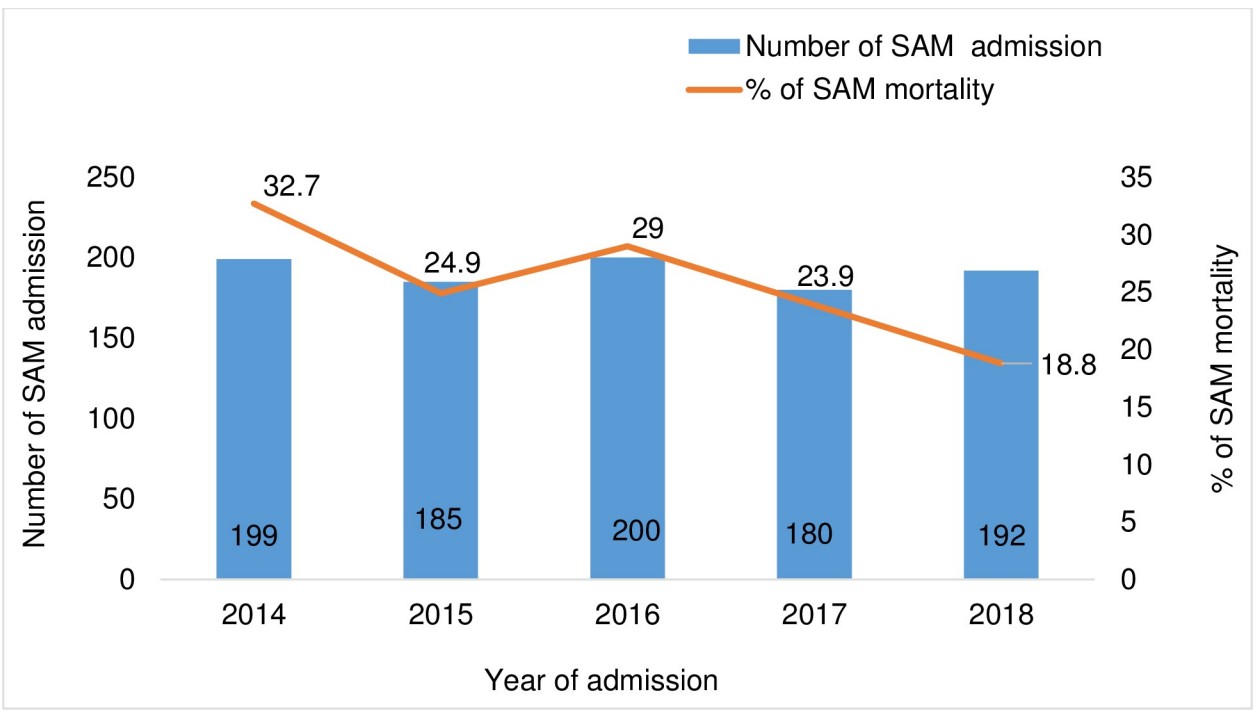

**Fig 2. Trends of SAM mortality in children under five years by year and number of admission at the study sites, 2014–2018.**

discharge and the median length of hospital stay was eight days (IQR: 5 to 14 days). The number of children admitted with SAM over the study period showed little variation (Fig 2). Apart from a slight increase between 2015 and 2016, mortality declined from 32.7% in 2014 to 18.8% in 2018. None of the children admitted at the study sites was transferred to other hospitals.

### Factors associated with SAM mortality

After adjusting for other variables, the odds of dying in children with MUAC <11.5 cm were three times higher compared to those who had a MUAC ≥11.5 cm (95% CI 1.9–4.7, p<0.001) (Table 5). Using herbal medicine and having poor appetite on admission increased the odds of dying by more than two fold (95% CI 1.4–3.5, p = 0.001) and (95% CI 1.4–5.2, p = 0.003) respectively. Children who had LRTIs had 1.6 times higher odds of dying compared to those who did not have LRTIs (95% CI 1.3–2.0, p<0.001). Children with anemia had 2.5 (95% CI 1.1–5.3, p = 0.021) greater odds of mortality while those with hypoglycemia had12.5 (95% CI 7.1–21.8, p<0.001) odds of dying compared to those who did not have these conditions. In addition, HIV positivity increased the odds of mortality by 2.3 times (95% CI 1.6–3.3, p = 0.016).

### Discussion

Our study aimed to identify factors associated with mortality among children under five years admitted with SAM in public hospitals of Limpopo province. We found an overall mortality of 25.9% among children admitted with SAM. Factors associated with mortality included MUAC <11.5 cm, poor appetite on admission, history of herbal medication use, LRTIs, hypoglycemia and HIV infection.

**Table 5. Logistic regression analysis of factors associated with SAM mortality in children under five years at the study sites, 2014–2018.**

| Variable | SAM mortality n/N (%) | Univariable analysis | | Multivariable analysis | |
|---|---|---|---|---|---|
| | | OR (95% CI) | *P-value | aOR (95% CI) | P- value |
| **Age (months)** | | | | | |
| 25–60 | 31/130 (23.9) | Reference | - | | |
| 0–6 | 41/120 (34.2) | 1.7 (0.9–3.0) | 0.101 | | |
| 7–24 | 176/706 (24.9) | 1.1(0.6–1.7) | 0.819 | | |
| **Sex of children** | | | | | |
| Male | 122/480 (25.4) | Reference | - | | |
| Female | 126/476 (26.5) | 1.1(0.8–1.4) | 0.678 | | |
| **Admission type** | | | | | |
| New | 199/809 (24.6) | Reference | - | | |
| Readmission | 49/147 (33.3) | 1.5 (0.9–2.5) | 0.091 | | |
| **Oedema** | | | | | |
| Non-Oedematous | 144/588 (24.5) | Reference | - | | |
| Oedematous | 104/368 (28.3) | 1.2(1.0–1.4) | 0.033 | | |
| **MUAC (cm)** | | | | | |
| ≥11.5 | 34/278 (12.3) | Reference | - | - | |
| <11.5 | 165/483 (34.2) | 3.7(2.1–6.6) | <0.001 | 3.0(1.9–4.7) | <0.001 |
| **Weight for length Z-Scores (SD)** | | | | | |
| 0 to -3 | 54/307 (17.6) | Reference | - | | |
| <-3 | 154/551 (27.9) | 1.8(1.1–2.9) | 0.013 | | |
| **Appetite** | | | | | |
| Good | 20/209 (9.6) | Reference | - | | |
| Poor | 188/654 (28.8) | 3.8(1.9–7.6) | <0.001 | 2.7(1.4–5.2) | 0.003 |
| **Immunization status** | | | | | |
| Up to date | 133/594 (22.4) | Reference | | | |
| Not Up to date | 77/281 (27.4) | 1.3(0.9–1.8) | 0.075 | | |
| **Feeding history** | | | | | |
| Adequate | 40/180 (22.2) | Reference | | | |
| Inadequate | 149/615 (24.2) | 1.1(0.6–1.9) | 0.701 | | |
| **History of herbal medication use** | | | | | |
| No | 170/788 (21.6) | Reference | - | - | - |
| Yes | 78/168 (46.4) | 3.2(2.1–4.8) | <0.001 | 2.2(1.4–3.5) | 0.001 |
| **Diarrhoea** | | | | | |
| No | 76/346 (21.9) | Reference | - | - | - |
| Yes | 172/610 (28.2) | 1.4(1.1–1.7) | 0.001 | | |
| **LRTIs** | | | | | |
| No | 123/551 (22.3) | Reference | - | | |
| Yes | 125/405 (30.9) | 1.6(1.1–2.2) | 0.014 | 1.6(1.3–2.0) | <0.001 |
| **Anaemia** | | | | | |
| No | 165/783 (21.1) | Reference | - | - | - |
| Yes | 83/173 (48.0) | 3.5(1.9–6.2) | <0.001 | 2.5(1.1–5.3) | <0.021 |
| **Hypoglycaemia** | | | | | |
| No | 171/860 (19.9) | Reference | - | - | - |
| Yes | 77/96 (80.2) | 16.3(8.6–31.1) | <0.001 | 12.5(7.1–21.8) | <0.001 |
| **TB** | | | | | |
| Not infected | 199/829 (24) | Reference | - | | |
| Infected | 49/127 (38.6) | 1.9(1.1–3.6) | 0.022 | | |

*(Continued)*

**Table 5.**  (Continued)

| Variable | SAM mortality n/N (%) | Univariable analysis | | Multivariable analysis | |
|---|---|---|---|---|---|
| | | OR (95% CI) | *P-value | aOR (95% CI) | P- value |
| HIV infection | | | | | |
| Not infected | 171/775 (22.1) | Reference | - | - | - |
| Infected | 77/181 (42.5) | 2.6(1.9–3.5) | <0.001 | 2.3(1.6–3.3) | 0.016 |

OR = Odds ratio. aOR = Adjusted odds ratio. CI = confidence interval. MUAC = Mid Upper Arm Circumference. SD = Standard Deviation. LRTIs = Lower respiratory tract infections. TB = Tuberculosis. HIV = Human Immunodeficiency Virus. *P- value for univariable analysis. P-value for multivariable analysis. P- value significant at ≤0.05.

We found that mortality among children with SAM in Limpopo was higher than the target outlined in the WHO SAM management guidelines (<10%) [1]. Mortality among children under five years with SAM in our study (25.9%) is comparable with figures reported in health facilities in rural Eastern Cape province (24.4%) [21]. This increased mortality could be due to delays in seeking treatment in the hospitals, leading to complications [21]. Nonetheless, mortality in our study was lower than figures reported in Zambia (46%) [12], Malawi (42%) [11] and Swaziland (40.1%) [23]. The observed differences between our findings could be due to differences in severity of cases, and frequency of co-morbidities [24, 25]. Mortality trends varied between 2014 and 2016; however, a gradual decline ensued from 2016 to 2018. Given the fairly consistent number of SAM admissions from 2014 to 2018 in our study, the decrease in SAM mortality from 2016 to 2018 is particularly encouraging. The observed pattern of mortality in our study is comparable to that reported by Bamford et al [20] and the District Health Barometer [17] in Limpopo province from 2014 to 2018. Variations in mortality over time were also observed in studies done in Ethiopia [13] and Zambia [12]. Numerous factors and interventions implemented in addition to WHO SAM management guidelines could explain the decline in mortality. These factors include; a decrease in HIV incidence and prevalence in young children following scaling up of prevention of mother-to-child transmission interventions, [17, 22] and improvements in key child health strategies such as immunization and exclusive breastfeeding [17, 26]. In addition, ward based outreach team members for assessment of SAM within the communities were established and trained from 2015 [27]. This might have resulted in early diagnosis and presentation of SAM cases in hospitals before they developed severe complications [28]. Munthali argued that the introduction of numerous interventions in addition to implementation of WHO SAM management guidelines have resulted in decreased mortality among children with SAM in Zambia [12].

Regarding predictors of mortality, history of using herbal medication, poor appetite on admission, MUAC <11.5 cm, LRTIs, anemia, hypoglycemia and HIV infection were associated with increased mortality. The use of herbal remedies in our study setting was consistent with findings of previous studies that reported caregivers administering herbal medicines to their children [29, 30]. Consultations with practitioners who administer herbal medicines may delay presentation of cases in health facilities, thereby increasing the risk of complications. Herbal medicines may also cause potential fatal complications such as diarrhoea, dehydration, liver and kidney impairments [31]. Management of liver and kidney impairment in children with SAM is not included in the WHO SAM management guidelines [1]. Treating children with SAM with complications that are not included on the management guidelines may be challenging to healthcare workers, and result in poor treatment outcomes. Our finding regarding the association between anemia and increased mortality in children with SAM reflect those of studies done in Sekota and Jimma Hospitals in Ethiopia [13, 25, 32]. Children with anemia

have increased risk of infections and heart failure, which may lead to fatal outcomes. Concerning LRTIs, hypoglycaemia and HIV infection, our findings agree with retrospective cohort studies done in Zambia and Ethiopia [12, 16]. Children with hypoglycemia have increased risk of neurological damage [33] while HIV infected children are predisposed to opportunistic infections [4, 33]. Tuberculosis was not associated with increased mortality in our study. However, this association between TB and mortality in children with SAM varies between studies with some finding no association, [25] while others did find an association [24]. These differences might be due to variability in adherence to WHO SAM treatment guidelines and early TB diagnosis as well as the provision of prophylaxis. Other factors explored in our study were poor appetite and MUAC <11.5 cm. The associations of MUAC <11.5 cm and SAM mortality was also reported in studies done in India [34] and Guinea-Bissau [35].

Our study had some limitations. Firstly, we included records of children with SAM who died at the study sites. This approach does not account for community deaths so it is not clear if factors are common to both groups. Secondly, data on some important variables were missing in included records. We addressed possible bias caused by missing information through multiple imputation. Due to the fact that our study entailed extracting data from hospital records, we could only include records that were available at the hospitals which did not have electronic archives. However, we have no reason to think that the characteristics of children with missing records differed from those that we obtained. We can outline major strengths of the study. We conducted a multicenter study including regional and district public hospitals found in two districts of Limpopo province. As such, our findings could be generalized to children under five years old making use of public sector health facilities of Limpopo province.

## Conclusions

Mortality among children with SAM was higher than the minimum standard of the WHO SAM management guidelines throughout the study period. History of using herbal medicines, lack of appetite, MUAC <11.5 cm, LRTIs, anemia, hypoglycemia and HIV infection were factors associated with increased mortality. These factors should inform management of children with SAM. There is a need to enhance community level interventions, targeted towards awareness of rational use of herbal medicines. Future studies assessing factors for mortality in children with SAM could be community-based to fill the knowledge gaps not addressed by our study.

## Supporting information

**S1 File. Ethical approval letter.**
(PDF)

**S1 Dataset. Manuscript data.**
(XLSX)

## Author Contributions

**Conceptualization:** Fhatuwani Gavhi, Lazarus Kuonza, Nkengafac Villyen Motaze.

**Data curation:** Fhatuwani Gavhi.

**Formal analysis:** Fhatuwani Gavhi, Alfred Musekiwa, Nkengafac Villyen Motaze.

**Methodology:** Fhatuwani Gavhi, Lazarus Kuonza, Nkengafac Villyen Motaze.

**Project administration:** Fhatuwani Gavhi.

**Supervision:** Lazarus Kuonza, Nkengafac Villyen Motaze.

**Writing – original draft:** Fhatuwani Gavhi.

**Writing – review & editing:** Fhatuwani Gavhi, Lazarus Kuonza, Alfred Musekiwa, Nkengafac Villyen Motaze.

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
