## [Decision Letter · Decision Letter 0]

8 Jan 2020

PONE-D-19-31811

Risk factors associated with mortality in children under five years old hospitalized for Severe Acute Malnutrition in Limpopo province, South Africa, 2014-2018: A cross-sectional analytic study

PLOS ONE

Dear Ms. Gavhi,

Thank you for submitting your manuscript to PLOS ONE. After careful consideration, we feel that it has merit but does not fully meet PLOS ONE’s publication criteria as it currently stands. Therefore, we invite you to submit a revised version of the manuscript that addresses the points raised during the review process.

We would appreciate receiving your revised manuscript by Feb 22 2020 11:59PM. To enhance the reproducibility of your results, we recommend that if applicable you deposit your laboratory protocols in protocols.io, where a protocol can be assigned its own identifier (DOI) such that it can be cited independently in the future. For instructions see: http://journals.plos.org/plosone/s/submission-guidelines#loc-laboratory-protocols

We look forward to receiving your revised manuscript.

Kind regards,

Calistus Wilunda, DrPH

Academic Editor

PLOS ONE

Journal Requirements:

Additional Editor Comments (if provided):

Consider using “Factors associated with mortality…” instead of “Risk factors associated with mortality…” in the heading and elsewhere in the manuscript.

Consider using methods such as multiple imputation to account for missing data. This can be done as a sensitivity analysis.

Line 298: The conclusion about representativeness may not be true? Did you check for representativeness? Exclusion of children with missing hospital records and restriction of the study to children admitted in public hospitals could have affected representativeness.

Reviewers' comments:

Reviewer's Responses to Questions

**Comments to the Author**

1. Is the manuscript technically sound, and do the data support the conclusions?

Reviewer #1: Yes

Reviewer #2: Partly

2. Has the statistical analysis been performed appropriately and rigorously? 

Reviewer #1: Yes

Reviewer #2: No

3. Have the authors made all data underlying the findings in their manuscript fully available?

Reviewer #1: Yes

Reviewer #2: Yes

4. Is the manuscript presented in an intelligible fashion and written in standard English?

Reviewer #1: Yes

Reviewer #2: Yes

5. Review Comments to the Author

Reviewer #1: I have read with interest this paper. It is a very interesting study done in the province of Limpopo in South Africa. This cross-sectional analytic study was able to research the risk factors for mortality in children under 5 years of age with severe acute malnutrition in whom the mortality rate was 25.9%. The risk factors found in this study have also been reported by other studies elsewhere.

We made some remarks:

Introduction section :

In your introduction, we note that you did not state a clear research question. Please could you include a statement regarding your research question in the introduction.

Materials and methods section :

It would be better to put references to these operational definitions.

Please cite co-morbidities and complications of SAM in this section.

Reviewer #2: The study by Gavhi, Kuonza, and Motaze describes risk factors for mortality in children aged under 5 years admitted with severe acute malnutrition (SAM). Participants were identified from admission registers of 7 participating hospitals followed by retrospective records review for variables of interest. Authors then performed logistic regression to identify risk factors for in-hospital mortality. Malnutrition remains a risk factor for both in-hospital and community mortality in children aged less than five year and case fatality from SAM remains high despite recommendations that have been there for about 2 decades. The topic is therefore of interest to scientists, clinicians and policy makers.

Major comments;

• The study uses retrospective review of routine medical records and therefore problems often encountered when dealing with routine data emerge as shown by high level of missingness in variables included in the analysis. Without understanding the pattern of missingness of the variables (MAR, MCAR, MNAR) then it is difficult to know if it is appropriate to do complete case analysis as done by the authors in the models. Biased estimates can be obtained when missingness is informative. The authors should therefore investigate the nature of missingness and convince readers that it is appropriate to perform complete case analysis without need to account for missing data in their analysis

• Data was collected from 7 hospitals (therefore clustered) but the analysis does not account for the clustered nature of the data. Multilevel modelling would be more appropriate

• Authors collected information on treatment variables in addition to other exposure variables however the analysis does not describe how the treatment variables were analysed when identifying risk factors. Risk factors as presently described have been previously described and what would be more informative is if an analysis incorporating the effect modification of WHO recommended treatment as decribed by Knol MJ (Int J Epidemiol 2012) was performed. Can the authors also include information on appropriate feeds as part of treatment information

• Was definition of SAM based on clinician diagnosis only (rather than algorithm defined); if yes then there is potential for missing cases and this affects both numerators and denominators

Minor comments

• Lines 112-114; The age range here is large and older children may be adequately fed on solid foods without need for formula supplementation or breastfeeding

• Line 116: Assumes that causality can be ascertained-rephrase

• Line 197-rehydration therapy is recommended in children with diarrhoea; denominator should be children with diarrhoea

6. PLOS authors have the option to publish the peer review history of their article (what does this mean?). If published, this will include your full peer review and any attached files.

Reviewer #1: Yes: Olivier Mukuku

Reviewer #2: Yes: Samuel Akech

---

## [Author Response · Author response to Decision Letter 0]

4 Mar 2020

Editor comments:

Comment: Journal requirements 

Response:Thank you for the comment, we amended accordingly. 

Comment: You indicated that you had ethical approval for your study. In your Methods section, please ensure you have also stated whether you obtained consent from parents or guardians of the minors included in the study or whether the research ethics committee or IRB specifically waived the need for their consent. 

Response: Thank you very much for the input. Ethics committee waived the need of the informed consent of the caregivers. We added this under method section. We also included ethics approval letter under supplementary materials. 

Comment: Please include captions for your Supporting Information files at the end of your manuscript, and update any in-text citations to match accordingly. 

Response: We added captions for ethics approval letter and dataset accordingly. We also updated in-text citation under ethical considerations (line 163).

Comment: Consider using “Factors associated with mortality…” instead of “Risk factors associated with mortality…” in the heading and elsewhere in the manuscript. 

Response: We amended accordingly. We used “Factors associated with mortality…” instead of “Risk factors associated with mortality”.

Comment: Consider using methods such as multiple imputation to account for missing data. This can be done as a sensitivity analysis. 

Response: We did multiple imputation to account for missing data. 

Comment: Line 298: The conclusion about representativeness may not be true? Did you check for representativeness? Exclusion of children with missing hospital records and restriction of the study to children admitted in public hospitals could have affected representativeness. 

Response: We amended the sentence to reflect that our findings apply to children admitted in public health sector facilities. We assume that this implies our findings do not apply to children who are not admitted and those who make use of private sector facilities. Regarding missing records, we do not think that the characteristics of children with missing records differed from those that we obtained.

Reviewer 1 comments:

Comment: Introduction section;

In your introduction, we note that you did not state a clear research question. Please could you include a statement regarding your research question in the introduction. 

Response: Thank you for the comment. In the last paragraph of the introduction, we provided a justification for the study and stated the aim of this study.

Comment: Materials and methods section;

It would be better to put references to these operational definitions.

Please cite co-morbidities and complications of SAM in this section.

Response: Thank you for the feedback. The terms complication and co-morbidities have been used extensively in the literature on SAM. In order to avoid confusion, we decided to split them in the manner that we did. In that way, we can reasonably distinguish medical conditions that were present in the patient before development of SAM from conditions that occurred as a result of SAM.

Reviewer 2 comments

Major comments: 

The study uses retrospective review of routine medical records and therefore problems often encountered when dealing with routine data emerge as shown by high level of missingness in variables included in the analysis. Without understanding the pattern of missingness of the variables (MAR, MCAR, MNAR) then it is difficult to know if it is appropriate to do complete case analysis as done by the authors in the models. Biased estimates can be obtained when missingness is informative. The authors should therefore investigate the nature of missingness and convince readers that it is appropriate to perform complete case analysis without need to account for missing data in their analysis 

Response: Thank you for the comment. Our data was missing at random (MAR). Therefore, we did multiple imputation and reported final model of imputed data.

Comment: Data was collected from 7 hospitals (therefore clustered) but the analysis does not account for the clustered nature of the data. Multilevel modelling would be more appropriate. 

Response: We revised accordingly by adjusting for clusters in univariable analysis and in our final model. 

Comment: Was definition of SAM based on clinician diagnosis only (rather than algorithm defined); if yes then there is potential for missing cases and this affects both numerators and denominators. 

Response: This study was based on collecting data from medical records and we included only children who were diagnosed with SAM by the treating clinician.

Comment: Authors collected information on treatment variables in addition to other exposure variables however; the analysis does not describe how the treatment variables were analyzed when identifying risk factors. 

Response: We limited ourselves to describing the type of treatment received, as we did not collect information of all SAM treatment regimens.

Comment: Risk factors as presently described have been previously described and what would be more informative is if an analysis incorporating the effect modification of WHO recommended treatment as decribed by Knol MJ (Int J Epidemiol 2012) was performed. 

Response: It is true that several studies have addressed risk factors for mortality in children with SAM. However, in our study, we were interested in exploring risk factors in a specific province of South Africa. We did not collect enough data on WHO recommended treatment to enable exploration of effect modification and so we are unable to perform this analysis. 

Comment: Can the authors also include information on appropriate feeds as part of treatment information. 

Response: We did not collect information on appropriate feeds during hospital admission. 

Minor comments 

Lines 112-114; The age range here is large and older children may be adequately fed on solid foods without need for formula supplementation or breastfeeding. 

Response: We amended accordingly. We divided age category 7 to 59 months into two categories (7 to 24 months and 25 to 59 months). 

Comment: Line 116: Assumes that causality can be ascertained-rephrase. 

Response: We rephrased accordingly. We wrote “Any medical condition occurring in a child with SAM” instead of “Any medical condition occurring because of SAM”.

Comment: Line 197-rehydration therapy is recommended in children with diarrhoea; denominator should be children with diarrhoea. 

Response: Thanks for the comment. Extensive literature on SAM has demonstrated that all children with SAM have some degree of dehydration, which is why rehydration is an obligatory component when treating children with SAM. Furthermore, dehydration can be due to other causes other such as vomiting or inadequate fluid intake- due to loss of appetite.

---

## [Decision Letter · Decision Letter 1]

23 Apr 2020

Risk factors associated with mortality in children under five years old hospitalized for Severe Acute Malnutrition in Limpopo province, South Africa, 2014-2018: A cross-sectional analytic study

PONE-D-19-31811R1

Dear Dr. Gavhi,

We are pleased to inform you that your manuscript has been judged scientifically suitable for publication and will be formally accepted for publication once it complies with all outstanding technical requirements.

With kind regards,

Calistus Wilunda, DrPH

Academic Editor

PLOS ONE

Additional Editor Comments (optional):

Reviewers' comments:

Reviewer's Responses to Questions

**Comments to the Author**

1. If the authors have adequately addressed your comments raised in a previous round of review and you feel that this manuscript is now acceptable for publication, you may indicate that here to bypass the “Comments to the Author” section, enter your conflict of interest statement in the “Confidential to Editor” section, and submit your "Accept" recommendation.

Reviewer #1: All comments have been addressed

Reviewer #2: All comments have been addressed

2. Is the manuscript technically sound, and do the data support the conclusions?

Reviewer #1: Yes

Reviewer #2: Yes

3. Has the statistical analysis been performed appropriately and rigorously? 

Reviewer #1: Yes

Reviewer #2: Yes

4. Have the authors made all data underlying the findings in their manuscript fully available?

Reviewer #1: Yes

Reviewer #2: Yes

5. Is the manuscript presented in an intelligible fashion and written in standard English?

Reviewer #1: Yes

Reviewer #2: Yes

6. Review Comments to the Author

Reviewer #1: The authors have appropriately addressed prior suggestions. Otherwise, I think the manuscript is well-revised.

Reviewer #2: The authors have adequately addressed previous comments

7. PLOS authors have the option to publish the peer review history of their article (what does this mean?). If published, this will include your full peer review and any attached files.

Reviewer #1: Yes: Olivier Mukuku

Reviewer #2: Yes: Samuel Akech

---

## [Editor Report · Acceptance letter]

28 Apr 2020

PONE-D-19-31811R1 

Factors associated with mortality in children under five years old hospitalized for severe acute malnutrition in Limpopo province, South Africa, 2014-2018: A cross-sectional analytic study 

Dear Dr. Gavhi:

I am pleased to inform you that your manuscript has been deemed suitable for publication in PLOS ONE. Congratulations! Your manuscript is now with our production department. 

With kind regards,

on behalf of

Dr. Calistus Wilunda 

Academic Editor

PLOS ONE